# PrEP knowledge, acceptability, and implementation in Ghana: Perspectives of HIV service providers and MSM, trans women, and gender diverse individuals living with HIV

Akua O. Gyamerah[1¤a]*, Ezra Kinzer[1], Gloria Aidoo-Frimpong[2¤b], Guro Sorensen[3], Matilda D. Mensah[3], Kelly D. Taylor[4], Naa Ashiley Vanderpuye[3], Sheri A. Lippman[1]

1 Center for AIDS Prevention Studies, Department of Medicine, University of California, San Francisco, San Francisco, California, United States of America, 2 Department of Community Health and Health Behavior, The State University of New York at Buffalo, Buffalo, New York, United States of America, 3 West Africa AIDS Foundation, Accra, Ghana, 4 Institute for Global Health Sciences, University of California, San Francisco, San Francisco, California, United States of America

¤a Current address: Department of Community Health and Health Behavior, The State University of New York at Buffalo, Buffalo, New York, United States of America
¤b Current address: Center for Interdisciplinary Research on AIDS, Yale University, New Haven, Connecticut, United States of America
* akuagyam@buffalo.edu

**Data Availability Statement:** The data used for the manuscript has been deposited and can be

## Abstract

Pre-exposure prophylaxis (PrEP) could help reduce HIV incidence among cis men, trans women, and gender diverse individuals assigned male at birth who have sex with men (MSM, trans women, and GDSM) in Ghana, a group that bears a high HIV burden. Our study examined PrEP knowledge and acceptability, and barriers and facilitators to its uptake and implementation through qualitative interviews with 32 MSM, trans women, and GDSM clients living with HIV, 14 service providers (SPs), and four key informants (KIs) in Accra, Ghana. We interviewed participants about their PrEP knowledge, whether MSM would take PrEP, and what factors would make it easy/difficult to uptake or implement PrEP. Interview transcripts were analyzed using thematic analysis. There was high acceptability of PrEP use and implementation among MSM, trans women, GDSM, and SPs/KIs in Ghana. MSM, trans women, and GDSM interest in, access to, and use of PrEP were shaped by intersectional HIV and anti-gay stigma; PrEP affordability, acceptability, and ease of use (e.g., consumption and side effects); sexual preferences (e.g., condomless sex vs. condom use), and HIV risk perception. Concerns raised about barriers and facilitators of PrEP use and implementation ranged from medical concerns (e.g., STIs; drug resistance); social behavioral concerns (e.g., stigma, risk compensation, adherence issues); and structural barriers (e.g., cost/affordability, govern commitment, monitoring systems, policy guidance). Targeted education on PrEP and proper use of it is needed to generate demand and dispel worries of side effects among MSM, trans women, and GDSM. Free, confidential, and easy access to PrEP must be supported by health systems strengthening, clear prescription guidelines, and anti-stigma training for providers.

accessed by reviewers during the peer review process. The data however, cannot be made public due to ethics agreement detailed in our consent form to keep data confidential and due to the context of criminalization and backlash against LGBTQI+ people in Ghana. In 2021, an anti-LGBTQI bill was proposed in Ghana's Parliament that would further criminalize LGBTQI people, advocacy, organizations, and allyship. Parliament is currently debating this bill and may vote on it this year. During the same year, the police also raided several LGBTQI organizations, events, and spaces. These ongoing attacks have made Ghana even more unsafe for LGBTQI Ghanaians and as researchers, we want to be cautious about making publicly available study data that may be used to fuel more attacks in Ghana. Questions about manuscript data can be sent to Akuagyam@Buffalo.edu or IRB@ucsf.edu. Below is an excerpt from our consent form on confidentiality and who has access to the study data. As reflected in the consent form, publishers and the general public were not included in the list of potential groups that can access study data.

**Funding:** This research was financially supported by a grant from the National Institutes of Health, University of California, San Francisco, Center for AIDS Prevention Studies (2P30MH062246) awarded to AOG. This work was also financially supported, in part, by a training grant from the National Institute of Mental Health (T32 MH19105–28) and a CFAR-ARI Boost Award from the UCSF-Gladstone Center for AIDS Research (NIH P30AI027763) awarded to AOG. The funders had no role in study design, data collection and analysis, decision to publish, or preparation of the manuscript.

**Competing interests:** The authors have declared that no competing interests exist.

# Introduction

Men who have sex with men (MSM) in sub-Saharan Africa (SSA) bear a disproportionate burden of HIV, with an estimated prevalence of 13.7% compared to prevalence of 6.7% and 1.4% among the general adult population in eastern and southern African and west and central Africa, respectively [1]. Among trans women in SSA, HIV prevalence is estimated at 25% [2]. Various factors place African MSM and trans women at risk of HIV, including anti-gay criminalization, gender and sexuality-based stigma and discrimination, and lack of access to tailored prevention services [2–4]. To address these disparities, comprehensive HIV prevention interventions are needed to help reduce HIV incidence. Pre-exposure prophylaxis (PrEP)—an effective HIV prevention medication—could help reduce HIV incidence among African cis men, trans women, and gender diverse individuals assigned male at birth who have sex with men (MSM, trans women, and GDSM) by up to 44% when used properly [5]. However, like many other regions, most African countries have yet to include it in their prevention strategy or have been slow to start up programming due to various factors, including health systems challenges, funding, and political will [6].

In Ghana, compared to the overall HIV prevalence of 1.7%, MSM have the highest HIV burden with prevalence at 18% [1]. There are no known estimates for trans women and GDSM. The high prevalence exists within a context where same-sex sexual activities are criminalized under a British colonial-era law [7]. Socio-structural factors such as stigma, discrimination, and violence have been associated with poor HIV prevention outcomes [8–10]. HIV and anti-gay stigma is particularly high and associated with lower condom use and HIV testing rates [8, 9, 11], which contributes to the context of high risk. Additionally, 48.2% of MSM nationally report always using a condom during anal intercourse with another man and 29.7% report always using a condom during sexual intercourse with a female partner [12]. About a third of MSM surveyed in four major regions of Ghana had tested for HIV in the preceding twelve months [10]. Given the high burden of HIV and low condom use and HIV testing, PrEP is a critically needed HIV prevention intervention for MSM, trans women, and GDSM in Ghana.

African MSM, trans women, and GDSM's PrEP interest and needs are not well understood, with most PrEP research in Africa predominantly focused on adolescent girls and young women and female sex workers [13]. However, a growing number of studies examining PrEP acceptability and use among MSM and trans women in African countries [14–22] suggest that PrEP knowledge is low but interest in PrEP use is high. Factors that shape PrEP interest include cost, anticipated or experienced stigma, ease of use, and history of experiencing sexual violence [14, 15, 17–20, 22, 23].

In addition, there is a gap in research on HIV service provider and stakeholder perspectives on PrEP uptake and implementation among MSM, trans women, and GDSM in Africa. Extant research in other low and middle income contexts has found that providers are interested in PrEP availability for MSM clients, but share concerns about risk compensation from PrEP use [24]. Provider views on PrEP uptake among trans women and GDSM are even less understood. Provider and stakeholder perspectives are especially important to understanding the health systems and other structural factors that may facilitate or hinder successful PrEP rollout, especially in under-resourced settings.

In August 2020, Ghana became one of a growing number of African nations to introduce PrEP to key populations, including MSM and later, trans women—a progressive step in the country's national HIV response [25]. While previous research on PrEP knowledge and acceptability among MSM in Ghana found low knowledge but high acceptability of PrEP once information about PrEP was provided [18], little is known about the views of trans women

and GDSM, as well as HIV service provider and stakeholder views on PrEP and what factors may facilitate or prevent access to PrEP if available. PrEP remains a prescription medication requiring the input and support of health system personnel. Effective implementation of PrEP in Ghana will partly depend on providers' perspectives on and willingness to prescribe PrEP.

To contribute to the growing literature on PrEP among African key populations, we nested questions about PrEP knowledge and acceptability into a study on the multilevel HIV treatment and care needs MSM living with HIV to explore client, provider, and stakeholder views on the potential for PrEP implementation and use in Ghana. As Ghana rolls out PrEP to key populations, more research is needed to better understand factors that shape PrEP uptake among MSM, trans women, and GDSM and provider perspectives to inform effective implementation for a key population facing high stigma and HIV prevalence.

## Materials and methods

### Study design and setting

The present research is part of a larger formative qualitative study examining the multilevel treatment and care needs of MSM living with HIV in Ghana, a West African country of 29.8 million people. The study was based in the Greater Accra Region where HIV prevalence among MSM is 42.2% (Ghana AIDS Commission, 2017). Ghana's epidemic is characterized as a low prevalence generalized epidemic mixed with a concentrated high prevalence epidemic among key populations, including MSM and trans women. In 2016, Ghana started implementing the universal Test and Treat policy developed by the World Health Organization in an effort to meet 90-90-90 treatment targets set by UNAIDS [26]. Despite this, only 34% of PLHIV are on ART [1]. The introduction of PrEP in 2020 represents an opportunity to provide highly efficacious prevention to key populations, however, roll out has been limited [25].

### Study population

Data collection occurred from March 2019 to July 2019. In-depth semi-structured interviews were conducted with a gender diverse group of 35 individuals assigned male at birth who have sex with men and with 14 service providers (SPs) and key informants (KIs), specifically case managers (CMs)/lay counselors (n = 8), nurses (n = 5), medical doctors (n = 1), and government and non-government key informants (n = 4). We also conducted one focus group interview with four MSM CMs/counselors.

**MSM, trans women, and GDSM living with HIV:** We used a combination of purposive and snowball sampling to recruit 35 MSM, trans women, and GDSM participants. Table 1 reports the sociodemographic characteristics of participants who were asked questions about PrEP (n = 32). MSM, trans women, and GDSM were approached by participating SPs and CMs from seven organizations and clinics that serve MSM clients. Eligibility criteria for MSM, trans women, and GDSM were: 1) assigned male at birth or self-identify as a man, 2) have male sexual partners, 3) HIV positive, 4) lives in Accra/Tema metropolitan area at time of enrollment, 5) age ≥18 years, and 6) speaks English. Participants were identified purposefully to represent diversity in terms of years living with HIV, HIV care engagement (being out of care or in different stages of the care continuum), ART adherence, and socioeconomic background. This ensured a range of perspectives on treatment and care experiences. Once a participant completed their interview, they were asked to refer up to two friends who fit the inclusion criteria. In total, 30 participants were recruited by providers and 5 were referred by a participating friend.

A note on gender identity within our sample. Although our study was aimed at understanding the treatment needs of MSM, we learned of the diversity of gender identities of study

**Table 1. Sociodemographic characteristics, knowledge, and acceptability of PrEP among MSM, trans women, and GDSM living with HIV (n = 32).**

| MSM participants | Age (years) (mean = 28.3) | Ethnicity | Employment status | Relationship status | Sexual orientation | Gender | Time since HIV diagnosis at study entry | Know of PrEP | Would have taken PrEP | Would other MSM take PrEP |
|---|---|---|---|---|---|---|---|---|---|---|
| P1[a] | 29 | Ga | Unemployed | Single | Gay | Male | <1 year | No | Yes | Yes |
| P2 | 25 | Ga | Unemployed | Single | Gay/MSM | Female | 2 years | Unsure | Yes | Yes |
| P3 | 25 | Frafra | Employed | Widowed | Gay | Female | 3 years | Yes | Yes | Unsure |
| P4 | 25 | Ga | Employed | Single | Gay | Female | 8 years | Yes | No | Unsure |
| P5 | 26 | Ga | Employed | Single | Gay | Male | 4 years | Yes | Yes | Unsure |
| P6 | 27 | Ga | Employed | Single | Gay | Male | 5 years | No | Yes | Yes |
| P7 | 31 | Ashanti | Employed | Single | Gay | Female | 11 years | No | Yes | No |
| P8 | 27 | Akan | Employed | Single | Gay | Female | 5 years | Yes | Yes | Unsure |
| P9 | 28 | Ga-Adangbe | Employed | In relationship (Female) | Bisexual | Male | 4 years | No | Yes | - |
| P10 | 27 | Ewe | Unemployed | Single | Gay | Trans woman | 2 years | No | Yes | Unsure |
| P12 | 50 | Kwahu | Employed | Single | Bisexual | Male | 2 years | No | Yes | Unsure |
| P14 | 30 | Ga/Fante | Employed | Single | Bisexual | Male | 1 year | No | Yes | Unsure |
| P15 | 21 | Fante | Unemployed | Single | Bisexual | Female | <1 year | No | Yes | Yes |
| P16 | 24 | Ewe/Fante | Training | Single | Bisexual | Both Male & Female | 1 year | Yes | Yes | Yes |
| P18 | 38 | Ewe | Employed | Single | Bisexual/Heterosexual | Male | 17 years | -[b] | No | - |
| P19 | 33 | Fante | Unemployed | Single | Gay | Male | 15 years | No | Yes | Yes |
| P20 | 26 | Ga | Employed | Single | Gay | Male | 1 year | Yes | Yes | Yes |
| P21 | 24 | Ga | Employed | Single | Bisexual | Male | 2 years | - | Yes | Yes |
| P22 | 20 | Ga | Employed | In a relationship | Gay | Male | 1 year | - | Yes | Yes |
| P23 | 27 | Akyem | Unemployed | Single | Gay | Female | 4 years | No | Yes | Yes |
| P24 | 25 | Ewe | Employed | Single | Bisexual | Both Male & Female | 7 years | - | Yes | Unsure |
| P25 | 29 | Guan | Employed | Single | Gay | Both Male & Female | 3 years | - | No | Unsure |
| P26 | 35 | Ga | Employed | Single | Gay | Both Male & Female | 4 years | - | Yes | Yes |
| P27 | 30 | Ga | Employed | Single | Bisexual | Male | 11 years | - | Yes | No |
| P28 | 24 | Fante | Employed | Single | Bisexual | Male | 2 years | - | Yes | Yes |
| P29 | 21 | Ga | Employed | Single | Gay | Male | 2 years | - | No | Unsure |
| P30 | 27 | Krobo | Unemployed | Single | Gay | Female | 4 years | Yes | Yes | Yes |
| P31 | 25 | Ga/Ewe | Unemployed | Single | Gay | Male | 1 year | - | Yes | Yes |
| P32 | 29 | Ga/Ewe | Employed | Single | Gay | Female | 4 years | No | Yes | Yes |
| P33 | 38 | Ga/Ashanti | Unemployed | Single | Straight | Trans woman | 10 years | Yes | Yes | Yes |
| P34 | 26 | Ga | Employed | Single | Gay | Male | 7 years | Yes | Yes | No |
| P35 | 34 | Ewe | Employed | Single | Gay | Male | 8 years | Yes | No | Yes |

[a] P = Participant

[b] "-" = not asked question or no answer

participants in the process of data collection. Specifically, when asked about gender identity, several respondents replied that they "feel like a woman" or feel like "both a man and a woman"; two people identified as transgender women. We subsequently included trans women and gender diverse individuals in the study due to how closely networked they and MSM are within Ghana's lesbian, gay, bisexual, transgender, and queer (LGBTQ) community and the fluidity of gender identity within the community. Additionally, the term "MSM" has become adopted as an identity among some people with diverse gender identities and expressions in the Ghanaian context. For example, some participants in our study identified as both MSM and female. We utilize the term cis men, trans women, and gender diverse individuals assigned male at birth who have sex with men (MSM, trans women, and GDSM) for reporting given this context, however, we note limitations to its use, including the conflation of gender identity and sexuality [27, 28]. Additionally, we occasionally use "MSM" as a descriptor when referring to questions in the interview guide which asked about the views, needs, and experiences of MSM related to PrEP.

**Service providers and key informants:** SPs (n = 14) and KIs (n = 4) represented local organizations that service MSM and LGBTQ communities, international donor agencies/bilateral institutions, international health agencies, and public and private clinics (Table 2). SPs were purposely selected from clinics that are designated as MSM-serving based on their experience working with PLHIV and MSM living with HIV in Ghana and willingness to participate in this study. Among SPs, seven CMs were MSM, and all were actively providing HIV prevention and care services to MSM clients. SPs were primarily trained as medical professionals or case managers, peer educators, and/or lay counselors. Key informants were selected based on organizational affiliations to ensure robust program representation, time and experience working with PLHIV and MSM within Ghana, and willingness to participate in this study.

**Table 2. Service provider and key informant characteristics and views on PrEP (n = 18).**

| PARTICIPANTS | Organization type | Position | PrEP Prioritization |
|---|---|---|---|
| KI 1 | Bilateral agency | Medical Doctor & NGO Project Director | Yes |
| KI 2 | Bilateral agency | Program Director | Yes |
| KI 3 | Government health agency | Medical Doctor/Public Health Director | Yes |
| KI 4 | University & teaching hospital | Researcher & Medical Doctor | Yes |
| CM 1 | LGBT rights NGO | NGO director/Lay Counselor | Yes |
| CM 2 | LGBT health NGO | Case Manager/Lay Counselor | No |
| CM 3 | LGBT health NGO | Case Manager/Lay Counselor | Yes |
| CM 4 | Public hospital | Case Manager/Helpline Counselor | Yes |
| CM 5 | HIV NGO | Case Manager/Lay Counselor | Yes |
| CM 6 | LGBT health NGO | Case Manager/Lay Counselor | Yes |
| CM 7 | International health agency | Case Manager/Lay Counselor | Yes |
| CM 8 | HIV NGO | Lay counselor | Yes |
| SP 1 | NGO/private clinic | Medical doctor | Yes |
| SP 2 | NGO/private clinic | Nurse | Yes |
| SP 3 | Public hospital | Nurse | No |
| SP 4 | Public teaching hospital | Nurse | Yes |
| SP 5 | Public clinic | Midwife/HIV & STI Counselor | No |
| SP 6 | Public clinic | Nurse | Yes |

*KI = Key informant; CM = Case manager; SP = Service provider

## Study procedures

When eligible participants were identified and consented to sharing their contact with study staff, they were contacted using a telephone/text script to schedule an interview. Interviews occurred at a private office at the study site in Accra and lasted between 45 to 120 minutes. Participants were reimbursed 50 cedis (~USD $10) for their time and transportation. Upon completion of interviews, interviewers asked participants to identify and reach out to one to two people in their network who meet the study criteria and might be interested in participating in the study. Referrals were not incentivized.

Interview guides for MSM, trans women, and GDSM participants assessed socioeconomic background, HIV history and treatment and care experiences, stigma and discrimination experiences, and knowledge and acceptability of PrEP. SP and KI interview guides assessed providers' experiences providing HIV services to MSM clients, views on HIV treatment policy guidelines, challenges and facilitators of service provision, PrEP need and implementation among MSM, and recommendations for addressing prevention and treatment needs of MSM. Out of the total sample of 35 MSM, trans women, and GDSM participants, 32 were asked questions about PrEP and all 18 SPs and KIs were asked about PrEP. PrEP questions explored PrEP knowledge (Pre-exposure prophylaxis is a prevention medication that people at high risk of HIV can take every day to prevent them from getting HIV. What do you know about it?), acceptability (If there was a daily pill you could take to prevent HIV, would you have taken it? Do you think other MSM you know would be interested in taking such a pill?), availability (Can you discuss whether there are any plans to introduce PrEP in Ghana?), and barriers and facilitators to its use and implementation (What would make it easy for you to take this pill every day?/What are the challenges/barriers to implementing PrEP in Ghana?).

## Data analysis

Interviews were conducted primarily in English, digitally recorded, and transcribed using third-party transcription services. Phrases in the transcript that were in Twi, a widely spoken local language, were translated to English by a member of the study team when needed. All interview transcripts were closely reviewed for accuracy and inconsistencies were corrected.

Transcripts were analyzed using thematic analysis [29]. An initial codebook was developed of a priori codes that captured the main in-depth and focus group interview guide themes, including a priori codes for PrEP questions. Two members of the study team conducted data analysis. To ensure reliability of analysis and findings, both (EK and AG) coded the first 15% of transcripts in the MSM and provider/stakeholder categories. The analysts then compared and discussed the coding, reconciled differences, and revised the codebook accordingly prior to the primary analyst (EK) completing coding of the remaining transcripts using the revised codebook. PrEP data segments were further analyzed for emergent themes and patterns and summarized for MSM and service providers/key informants respectively. Themes were displayed on a table for each group to compare similarities and differences between the two groups. Themes on PrEP use and implementation were then grouped under the following categories: 1) awareness of and interest in PrEP and its implementation; 2) reasons for PrEP interest; and 3) barriers and facilitators of PrEP use/implementation. Data analysis was conducted using Dedoose desktop and web-based version 8.3.35 [30]

## Ethical approval and consent to participate

The study received ethical approval from the Institutional Review Boards of the University of California, San Francisco and the Noguchi Memorial Institute for Medical Research at the University of Ghana, Legon. Before data collection commenced, participants were read an

informed consent form after which they selected a checkbox on the form to give consent to participate.

## Results

### Participant characteristics

There was a total of 32 MSM, trans women, and GDSM clients (Table 1) and 18 SPs and KIs (Table 2) in the present analysis. Mean age of clients was 28.3 years, with ages ranging from 20 to 56 years. Most clients identified as gay or bisexual, about one-third were unemployed, and all but two were single. At the time of the study, most clients (n = 23) had been diagnosed with HIV within the previous five years. Tables 1 and 2 further profile MSM and GDSM clients and SP/KI participants.

### Overview

Our findings are divided into two sections: 1) Client perspectives on PrEP, and 2) service provider and key informant perspectives on PrEP. Overall, there was great interest in PrEP use and availability for MSM, trans women, and GDSM among clients and SPs/KIs. MSM, trans women, and GDSM interest in, access to, and use of PrEP were shaped by intersectional HIV and anti-gay stigma; PrEP affordability and ease of use; sexual preferences; and HIV risk perception. Despite the resounding interest in PrEP, participants shared various multilevel concerns and factors that would shape PrEP use and implementation in Ghana for MSM, trans women, and GDSM. These ranged from medical concerns (e.g., STIs; drug resistance); social behavioral concerns (e.g., stigma, risk compensation, adherence issues); and structural barriers (e.g., cost/affordability, government commitment, monitoring systems, policy guidance). Table 3 displays these key themes and corresponding quotes from clients and SPs/KIs on PrEP knowledge, interest, and implementation for MSM, trans women, and GDSM. The sections below elaborate on these themes.

### Client perspectives on PrEP

**Awareness of PrEP.** Over half of clients who were asked about PrEP knowledge (n = 22) did not know of it (n = 12). Among those who were aware of PrEP (n = 10), some confused it with post-exposure prophylaxis (PEP). Those who reported they knew of PrEP had heard about it through other people/friends, social groups (e.g., an LGBTQ group), service providers, or on social media (e.g., Facebook pages) and other online platforms (e.g., YouTube). Once PrEP was described to participants, most (n = 27) expressed that they would have taken PrEP while they were HIV-negative.

**Reasons for PrEP interest.** Clients gave various reasons for interest in PrEP use and were related to sexual desires, cultures, and risk perception; social stigma; and desire to maintain their freedom and health. The majority of those who expressed interest in PrEP viewed it as a tool for investing in their health and maintaining control over sexual health and intimacy, particularly in social contexts where condom use is low. They also emphasized PrEP's potential to prevent a highly stigmatized disease like HIV and the added benefit of gaining freedom they associate with living without HIV. The minority of participants who were not interested in PrEP feared its use would stigmatize them as ill, feared PrEP side effects and health risks, preferred condom use over medication, and had low HIV risk perception.

**Table 3. PrEP interest and implementation themes from MSM, trans women, and GDSM client, service provider, and key informant interviews.**

| THEMES | QUOTES |
|---|---|
| **CLIENT INTEREST IN AND NEED OF PrEP** | |
| Interest in PrEP | CM 3: "A lot of people [have been] talking about [PrEP]. . .A lot of people come in and be like, do you have preexposure prophylaxis? I said no."—Case manager, LGBT health NGO |
| Reasons for wanting to take PrEP | P3: "Because I know that my husband likes raw sex. So, I would just take [PrEP] and be free to do my raw sex."—Gay, female, 25 years<br>P8: "I love being in a relationship. . .it'll get to a time you'd want to have sex with your partner and without even protecting yourself. And that is a high risk too."—Gay, female, 27 years<br>P12: "I'm tired, I'm a human being. Sometimes I feel sex, sometimes I feel it but because of [HIV] I don't want to do it. But the doctor told, the one who was taking care of me, he told me I can have sex because of the medicine that I'm taking. I said, 'But I don't feel it.' But sometimes I feel because I'm a human being."—Bisexual, male, 50 years |
| Reasons for not wanting to take PrEP | P25:" Because taking medicine. . .sometimes you get fed up. Why is it that always you're taking medicine?. . .People are gossiping. You yourself you don't even feel [good] because each time you have to take [pills] along with you. Maybe if you default one day, you end up [over]thinking. I wouldn't [take it] honestly."—Gay, male & female, 29 years |
| Priority groups in need of PrEP | KI 4: "So long as people don't want to always take [ART] and they want to be discordant couples. People say, 'this one I cannot do it with, if I put on the condom, I don't like [it].'"—Researcher & medical doctor, university & teaching hospital<br>SP 2: "I have one client, he always calls for PEP. And I asked him, 'PEP is meant for health workers. And so if you get exposed [to HIV] and I give it to you, we expect that you protect yourself.' But it keeps happening. . .and so you think that no, this guy needs PrEP."—Nurse, NGO/private clinic<br>SP 4 "If we can get funds to even run a project on adolescents on PrEP, we'll be glad, so that we see how it'll work for them. Because we have some adolescents that are concerned, they are sexually active. They can't disclose. Some of them have disclosed and they don't want the partner to be positive. So, they come to ask us because they read it on the net. Then we'll say that it's there but we don't have it."—Nurse, public teaching hospital |
| **MEDICAL CONCERNS** | |
| Distrust of medication/fear of side effects | P27: "If you don't know that you even have it and you are taking that medicine, you should know that everything is very bad. Taking medicine. . .has its own side effect. Taking in too much medicine can affect your systems because you'll be taking it and there's nothing wrong with you. When your system becomes weak like that, what will happen?"—Bisexual, male, 30 years |
| Ease of taking and adhering to PrEP pills | P28: "The weight of the medication and the design of the medication is bad. . .it would have been hard to take it persistent because people are not used to taking drugs oh! Because me, as I was growing up, the drugs I was comfortable taking was liquid drugs or something smaller. Even if it is tablet, but something smaller."—Bisexual, male, 24 years |
| ARV resistance | SP 5: "Some, when we introduce [PrEP] to them, some people will start. But somewhere along the line, they'll stop and they will get [HIV]. You understand? Some will take it and they would not take it well. And then maybe if the virus enters, they'll get resistance to [ARVs] and they have to change to a second one."—Midwife/HIV & STI counselor, public clinic |
| **SOCIAL BEHAVIORAL CONCERNS** | |
| PrEP/social stigma | SP 6: "The one major challenge would also be, they're scared they will be labeled when coming for the PrEP. And they would be interrogated or they will be chastised, the fear would also be there. The person may love it but wouldn't know how to come out."—Nurse, public clinic |

*(Continued)*

**Table 3.** (Continued)

| THEMES | QUOTES |
| --- | --- |
| Risk compensation and STI infections | CM 3: "When [PrEP] comes, we are just preventing HIV. What about the other STIs? There are lots of people that are suffering from HPV today or genital warts, anal warts today. . .Our people will go out and have unprotected sex and come back with an STI, different STIs altogether and then we'll be surprised to have other STIs coming that we hardly get in the system. . .because we will be doing all sorts of things."—Case manager, LGBT health NGO |
| Adherence to PrEP | CM 1: "It'll be a worry for MSM. . .to be taking drugs every day—a pill every day —unless they are diagnosed of something. Everything with that is a challenge—is a problem for them. I mean every individual doesn't want to be taking a pill every day, unless you are an addict. Yes. But most MSMs will not like to be taking pills every day."—Case manager, LGBT rights NGO |
| **STRUCTURAL BARRIERS AND FACILITATORS OF PREP USE AND IMPLMENTATION** | |
| Prep affordability and accessibility | P19: "If I'm working and I have money, for my protection, I will go for it. [For other MSM] it's all about if they have. If it's free, most of them will take it. But if it depends on money, it's a problem, because most of them don't work, that's the problem."—Gay, male, 33 years<br>CM 1: "Looking at the MSM community, [only] a few of the well-to-do MSM can afford it. But then, if the PrEP is supposed to be sold, we should forget it."—Case manager, LGBT rights NGO<br>P3:" Sometimes, even when you're going for the medication in the hospital, it's difficult. It's difficult because, sometimes you go there and then you meet someone that you know. And they know where we are taking our medication. . . When they are going there, people know that this is why you are going in there. Because that place is an emergency ward. And they know it."—Gay, female, 25 years |
| Government and donor commitment | CM 5: ". . .Looking at the bigger MSM community, they can't afford. So if it is to be subsidized, just like the government subsidized the antiretroviral, the people should be able to take it. And people will be willing to take it as long as they know that it will prevent HIV transmission and infection."—Case manager, HIV NGO |
| Health system challenges | KI 1: "The other concern is about tracking and making sure that those that are going to be on PrEP will be properly tracked, they keep their appointments, they will be doing their tests as required because if you're on PrEP and you seroconvert you have to be put on full treatment."—Medical doctor and director, bilateral agency |

## Control over sexual health and intimacy

Several clients discussed uptake of PrEP as a way to control and reduce risk during sexual activities like condomless sex, including situations where condom use is not possible or desired. Most participants were aware of their risk of acquiring or transmitting HIV due to sexual activities they or other MSM community members were engaged in. As a 20-year-old gay male participant (P22) explained, "Not everybody likes to use the condom". These perspectives on sexual preferences and practices among MSM and more generally in Ghana suggested that condom use is lower because condomless sex is considered more intimate, pleasurable, or convenient. For those who prefer "raw sex" and their partners, as well as in contexts where condoms are not accessible, PrEP would help offer protection against HIV transmission while maintaining the personal benefits of condomless sex. One participant (P20) explained:

> Within the saso [gay/MSM] community [and] within the heterosexual community, everyone finds it difficult to have protected sex. . .So if I know I'm the type who doesn't like to have protected sex, then I don't think anything should be preventing me from taking PrEP because at the end of the day, I would want to have sex without wearing a condom. So, it won't be difficult for me to take it knowing I'm sexually active and I would want to. . .save my life.—Gay, male, 26 years

Others described feeling restricted from meeting their human need for sexual intimacy due to their HIV-positive status. With PrEP, however, they would be able to be sexually intimate in ways they desire with partners without fear of HIV transmission (see quotes on Table 3).

Also related to control of sexual intimacy and health, some clients shared that PrEP would be useful for situations where a sex partner's HIV status is unknown. Notably, while clients stated desires for steady or exclusive relationships, some felt that the MSM community and social norms did not foster those types of relationships developing. With PrEP, they would not need to know their partner's HIV status or trust that their partner was using safer sex practices, "...If you think your partner is also into a promiscuous activity, you can also take [PrEP] to prevent yourself from [becoming] infected." Relatedly, some shared that PrEP would be useful in unpredictable situations, such as random sexual encounters and hookups or in case a condom breaks during sexual intercourse. As a 27-year-old gay female participant (P30) shared, "Of course, I will be taking [PrEP], because most definitely a hookup or something can come. And you know sometimes there are some sex that occur, you have no idea and before you know it, gbam! You understand?"

Notably, the few clients who were not interested in PrEP use also wanted to maintain sexual health and intimacy. However, they preferred to do so through condom use rather than taking PrEP because they found the idea of the PrEP regime—daily dosing—too demanding. Unlike those who preferred PrEP because of their dislike or skepticism of condoms, clients who preferred condoms found it easier to remember and more convenient to use than a daily medication. As a 21-year-old gay male participant (P29) shared, "...taking [PrEP] every day, I can't. I can't take medicine every day...I would feel stressed taking it every day...The medicine I know to prevent [HIV] is condom. It's not a medicine but that's what you get to protect yourself." Additionally, compared to those who expressed interest in PrEP due to high HIV risk perception, clients who were uninterested in PrEP shared that they would not have taken PrEP because they had no HIV risk perception before seroconverting. As a 26-year-old gay female client (P4) explained, "I wouldn't have taken it. Because I wasn't having any thoughts that people are having [HIV]. So, I didn't think I was high risk by then."

## Prevention of stigma

Another prominent theme that emerged from clients' discussion of PrEP interest was that of social stigmas associated with HIV and how these stigmas compromise their freedom. In particular, some participants viewed PrEP as a way of preventing a highly stigmatized disease like HIV, which they explained was associated with gay people in Ghanaian society. As one client (P26) shared,

*Society has been saying that because you're gay, you can be HIV positive. They think that is our punishment because you are gay that's why you are HIV positive. So why not take that medicine? I can take it, then I'll not be an HIV positive.—Gay, male & female, 25 years*

Another 25-year-old gay male client (P31) echoed the concerns about HIV stigma, stating that in Ghana, people view HIV as "a punishment" and that "discrimination for the sickness is very high". Thus, he believed MSM would be interested in taking PrEP because it would prevent the negative treatment associated with the disease, including social rejection and isolation. As a 29-year-old gay male participant (P1) explained, "In Ghana, when somebody hears you, 'Hey, this guy has HIV,' people won't come around you. People will reject you. You won't have a friend. You won't have anything to do with anybody...So if they will get [PrEP], that will be okay to prevent it."

Due to these intersecting stigmas, many participants shared that they would have taken PrEP to maintain their life, health, and freedom as an HIV-negative person. Some (P23 and P27) particularly emphasized a desire to have their "normal" or old life back, free from a life-changing illness. For these participants, falling "sick" with HIV meant losing their freedom. As a 27-year-old gay female participant (P23) explained, "When you are not sick, you have everything. So, if I would have known that this medicine was there, I would have taken it just to protect myself."

While those interested in PrEP viewed it as a way to prevent being stigmatized, those who would not have used PrEP feared that taking a daily pill would lead to possible stigma, suspicion, and gossip due to the association of pill dosing with illness.

## Investing in one's health

Finally, several people who expressed interest in PrEP regarded its use as an investment in their health. A 27-year-old gay female participant (P30) shared that she would take PrEP "...because my health is very prime to me, I don't joke with it." This view was echoed by others, who viewed PrEP as a way of maintaining strength (P1) and preventing early death (P2). For others, PrEP was worth investing money into for their health and that of their partners. For example, one participant (P30) explained that she would advise her partner to take PrEP and give him money to pay for it "because it's helping me". She also emphasized that she would even take the pills in front of her friends.

**Barriers and facilitators to PrEP use.** Participants discussed several factors that would facilitate or prevent PrEP uptake, regardless of their interest in taking it. Issues raised ranged from structural barriers such as accessibility and affordability to medical and social behavioral concerns related to PrEP such as ease of use, drug side effects/risks, and stigma. Notably, several of the factors discussed were similar to barriers participants were experiencing in taking antiretroviral medication.

## Structural concerns: PrEP affordability and accessibility

The issue of PrEP cost was commonly raised by participants when asked what would make it easier or harder to take PrEP. Since many of the participants were low-income or unemployed, affordability of PrEP was an important factor in decisions around PrEP. As a 25-year-old gay male client (P31) who had defaulted on ART explained, "It will be a financial problem because as of now, I'm unemployed. And the little money that I have, my parents have been giving me. So maybe the medicine will be very expensive. The difficult thing would be I can't afford it." Additionally, when asked about whether other MSM would take PrEP, a few participants shared that it would depend on the cost as many MSM do not have a stable source of income.

> *If it's expensive and people can't buy, it will make it difficult for them to take because, at the end of the day, I would want to have sex but then what it will take to prevent me from getting the virus is expensive, so I will however go in for the sex, but I won't take [PrEP]. (P30, 27-year-old, gay, female)*

Regardless of whether clients expressed interest in taking PrEP, many shared that taking PrEP would be difficult based on their current challenges with accessing ART. For them, easy, confidential accessibility would be critical to taking PrEP. For example, a 25-year-old gay female client (P3) who was diagnosed with HIV three years prior shared that getting HIV medication is difficult because of her fear of running into someone she know who might learn of their HIV status due to HIV stigma and had the same concern for accessing PrEP. Similarly,

due to social stigma she would be worried to run into a familiar face while picking up PrEP from the clinic.

## Medical and social behavioral concerns: ease, risks, and stigma of PrEP use

Other barriers related to challenges with ART use that clients raised about PrEP uptake was ease of use and adherence, as well as stigma associated with medication use. As a 26-year old gay male participant (P34) who had a history of defaulting from ART explained, some people believe that if "I'm taking a drug every day, it means I'm sick. So just that mentality alone doesn't really give way to having such medications like PrEP." Clients also discussed other concerns about daily dosing of PrEP, with some equating taking a daily pill with having HIV and others describing it as stressful, difficult, and tiring. Another 26-year old gay male client (P20) who was diagnosed a year prior explained that taking PrEP would be difficult "if the dosage is something I would have be taking on a daily basis" as if he is sick "because at the end of the day, I'm trying to prevent something and I'm living on [medication]." A 27-year-old gay female client (P23) additionally shared that people living with HIV "are having problems with-...taking drugs all the time", so she would have experienced the same pill fatigue if she took PrEP prior to seroconverting.

Other issues raised related to PrEP ease of use were PrEP medication route of administration and side effects/risks. Regarding administration, although we did not directly ask about injectable PrEP, some clients raised injection as a preference over taking a pill, while others wanted small, easy to consume pills. Additionally, several clients were worried about the possible health risks of taking a daily medication like PrEP over a long period of time. Unlike those who viewed PrEP as strengthening their health, a 30-year old bisexual male client (P27) spoke extensively about his concerns about medication side effects and its weakening effect on one's health. He explained that some people just do not trust that medications work as intended, so they would not take PrEP in fear that it will not protect them from contracting HIV. He added that some may fear of side effects like "dizziness".

## Service provider and key informant responses to PrEP

**Awareness of PrEP, its implementation, and ethical concerns.** All SPs/KPs were aware of PrEP, except for one CM who initially confused it with PEP. Three of 14 SPs did not think PrEP should be prioritized, whereas all KIs thought it should. Like MSM and GDSM clients, the majority wanted it to be available for key populations like MSM, serodiscordant couples, and people engaged in high-risk sexual activities and believed MSM clients would be interested in taking PrEP. Most SPs/KIs shared that PrEP was not yet publicly available in Ghana at the time of the interview and were not aware of government plans to make it available. A few KIs and SPs were, however, aware of plans to roll out PrEP to key populations in 2020. One government stakeholder (KI2) shared that donor and government stakeholders were discussing the introduction of PrEP to Ghana, however, they were first prioritizing linking PLHIV to ART to move more people closer to "undetectable = untransmittable (U = U)". Another KI (KI1) from an implementing partner organization shared that government and donor stakeholders were planning to rollout PrEP but there was skepticism about it and "a lot of concerns". He stated concerns about a lack of "safeguards" or "serious studies" to ensure the proper implementation of PrEP and felt the country had "a long way to go with PrEP".

MSM CMs were "ready" to have PrEP publicly available to their clients and did not understand why government and other stakeholders "just don't want to release it for people to take" as CM1 lamented. They expressed their frustration with the lag in PrEP rollout and asserted

that earlier introduction of PrEP in Ghana would have helped reduce HIV prevalence among MSM. As CM5 shared,

*I think [PrEP] is one thing that has wowed [helped] in a lot of places. And I think it has taken even too long for it to be rolled out here. . .If PrEP had been rolled out in maybe five years ago, the [HIV] prevalence as it stands now would have gone down among MSM. Because if I know I have a pill that if I take once a day, it stops transmission or prevents me from contracting HIV, why won't I take it?—Case manager, HIV NGO*

KI2 similarly expressed concern that the delayed introduction of PrEP was "becoming unethical" given the scale of the epidemic among MSM and knowledge that PrEP is effective in preventing HIV transmission.

*I feel it is unethical for me as a clinician [to] sit in the clinic to see key population clients. I feel that if. . .there is an MSM seronegative partner, why shouldn't I tell him or her that you can protect yourself from HIV infection because we have this medication?—Program Director, bilateral agency*

### Conflation of PrEP and PEP

One notable observation is that some providers and KIs made comments about PrEP that suggest that they might be confusing PrEP with PEP and indicate a need for further education on how PrEP works. For example, a few expressed the concern that PrEP would be misused routinely, drawing comparisons to the ways people use emergency contraceptives routinely. As KI2 shared, "The concern is using it as a routine rather than preventive. For example, we have emergency contraceptive, which is now being used as routine."

**Barriers and facilitators of PrEP implementation.** SPs and KIs expressed various challenges, considerations, and facilitators of PrEP rollout in Ghana. Concerns included 1) structural concerns like cost accessibility of the drug, health system challenges, and government/donor commitment to funding and sustaining PrEP, 2) medical concerns, like ARV resistance after seroconverting, and 3) social behavioral issues such as clients' ability to adhere to PrEP, stigma against PrEP users, abuse/misuse of PrEP, and risk compensation and STI infections.

### Structural concerns: PrEP affordability, government commitment and health system challenges

Similar to clients, SPs and KIs believed PrEP would be taken by MSM, but viewed the high cost of PrEP as a key challenge to its rollout in Ghana. One informant (KI2) referenced a U.S. Congressional House Committee hearing that occurred at the time of the interview during which Gilead was questioned on the high cost of PrEP. He explained that "PrEP cost $20,000 a year" per person in the U.S., and thus, he did not think it would come to Ghana "anytime soon". SPs additionally shared concerns that if PrEP were sold rather than highly subsidized or free, it may produce disparities among MSM in who can afford it.

Due to this high cost of PrEP, some SPs/KIs shared that government commitment to funding PrEP would be critical to its successful rollout and client access to it. A provider (SP3) believed it was possible to bring PrEP to Ghana, "if the government can sustain it". He referenced health system challenges such as antiretroviral shortages in Ghana and was worried that introducing PrEP would present "another burden". Similarly, a KI (KI1) shared that the country has been "having problems with sustainable levels of drugs for those infected [with HIV]"

so there is concern that introducing PrEP to Ghana would "compete with [HIV] treatment". He believed "political will" and "policy inertia" would be critical to addressing the various barriers to implementing PrEP in Ghana. Another KI (KI2), however, was concerned that the government may not be able to implement PrEP without donor funding and support.

Another health system-related challenge was that of monitoring and surveillance of people on PrEP, which KIs shared was a topic of discussion among stakeholders. Particularly, several stakeholders shared concerns about whether people on PrEP would be "properly tracked" to ensure timely identification of those who seroconvert so that they could be properly treated and reduce the risk of developing ARV resistance.

### Social behavioral concerns: adherence to PrEP

SPs and KIs gave a mix of responses when asked whether they believed MSM clients would take or adhere to taking PrEP medication. They made comparisons of PrEP to other HIV medications and expressed concern that MSM may experience similar barriers to adhering to PrEP as MSM living with HIV do with antiretrovirals. One MSM CM (CM4) explained, "there are still some [MSM] who wouldn't take it because even those who are positive that are supposed to take antiretroviral, they are not even taking it. How much more the PrEP?" He additionally shared that he and other MSM have debated whether PrEP "is going to work for us" due to concerns about side effects of other HIV medications like PEP, which he described as "unbearable". Another CM (CM5) made a similar point about medication side effects, ". . .people will have issues [taking PrEP]–I even know of health professionals who have been exposed [to HIV] taking PEP and some of them are not able to complete the therapy."

### Social behavioral concerns: Stigma from taking PrEP

Similar to concerns clients shared, some providers were worried that taking PrEP pills would be stigmatizing and a sign that clients are sick. One participant (CM1) expressed that taking the pill daily would be a challenge unless a client is diagnosed with an illness; it would also raise questions from people, "What is wrong with you? Are you sick? Why are you taking drugs every day? Are you abusing drugs or something?" Instead of daily pills, he suggested that a biannual injection would be a better prevention modality like some clients suggested. Some, however, did not think daily dosing would be an issue because taking PrEP would be a preventative measure rather than a reflection of illness.

Echoing the clients' concerns around intersectional stigma, providers shared that MSM might be worried about being outed or labeled as MSM if they visited the clinic for PrEP. As one nurse (SP6) explained, "The person may love [PrEP], but wouldn't know how to come out [for it]." A KI (KI4) shared similar concerns about stigma as a barrier to people seeking PrEP. They were particularly concerned that the stigmatizing healthcare environment may create a black market for PrEP, whereby some may sell the medication to those who don't want to appear at a clinic due to experienced or anticipated stigma, creating an issue with monitoring PrEP use. He explained, "There are people who would never go to the facility. If there are individuals who wouldn't know where to pick PrEP, [others] will be picking from nurses and selling them."

### Social behavioral concerns: Risk compensation and STI infections

Similar to clients, a few SPs and KIs raised concerns that clients may engage in riskier sexual behavior with the availability of PrEP. One MSM CM (CM2), for example, shared that he did not think PrEP should be available because he believed MSM would "abuse it". He was particularly concerned that some would have condomless sex with the intention of taking PrEP after,

"People would abuse it, especially MSM. As soon as they know that this one is there. He will go and do the raw sex and he will think that it will help him. So, he will come and take [PrEP]." He was also concerned that people would stop using condoms once enrolled on PrEP. However, he expressed that a strong educational component to the roll out would "encourage proper PrEP use". Another CM (CM3) shared his concern about MSM contracting STIs, especially HPV (anal warts), due to risk compensation once on PrEP. He expressed worry that MSM may forget that while PrEP protects against HIV, it does not prevent STIs.

## Medical concerns: drug resistance to ARVs

Several SPs and KIs raised concerns about clients possibly developing resistance to ARVs due to improper or sporadic use of PrEP. At the time of the study interviews, nearly all SPs and KIs had attended a recent HIV conference where PrEP was discussed, particularly the issue of patients developing resistance to ARVs. Thus, this concern was commonly raised during interviews. One provider (SP1), for example, was worried that those who seroconvert may develop resistance to ARVs and wanted this to be considered in the implementation of PrEP. She explained,

*We also need to think about what might come up after [starting PrEP] because when you are put on PrEP, if maybe it happens that you get infected, you can be resistant to that medication. So, I think it's good that we fight for the PrEP to be given, but we also need to think about what can come up later, about resistance, in case the person gets infected.—Medical doctor, NGO/private clinic*

Another provider (SP5) shared that people commonly start and stop taking medication. She was, thus, worried that those enrolled on PrEP would sporadically take the medication, get HIV, and develop resistance to ARVs due to intermittent use of PrEP. A medical doctor (KI4) also shared a concern that MSM who are already positive may knowingly enroll on PrEP either by lying or using the results of a negative person to bypass screening to PrEP and that this may cause them to develop resistance to ARVs.

## Desired features of PrEP implementation

SPs and KIs shared a few suggestions for the implementation of PrEP for MSM. Particularly, in addition to the call for government and donor commitment to implementing PrEP and subsidizing the cost, they emphasized mass and targeted education on PrEP with its rollout, what it does and does not protect against including understanding risks of STI, and proper use of it among MSM clients. Another provider lamented that local media outlets no longer broadcast ads or programs on HIV, which has reduced public knowledge of HIV, making it even more critical for PrEP to be implemented with a strong public education campaign. A CM (CM3) also noted that counseling of clients is an important feature of PrEP education and that implementers "should be ready to work hard", especially because some people think PrEP is a one-day pill. Another CM concerned about STI transmission shared that education among MSM on the importance of continuing condom use while on PrEP would be critical to rollout.

SPs and KIs also suggested that clear guidelines for providers on how to prescribe, counsel, and monitor PrEP use among MSM would be critical to PrEP implementation in Ghana. A KI (KI4) emphasized the need for clear policy guidelines for PrEP and to define the category of people who should receive PrEP to facilitate smoother implementation, "It's about how the country fashions it out and how they want to operationalize it." Others pointed out that implementation would require not only prescribing guidance, but also facilitated, stigma-free access.

To address the issue of PrEP and anti-gay stigma, some providers discussed the importance of offering PrEP through "key population friendly" clinics so that MSM clients will feel more comfortable seeking PrEP services. Another suggestion in response to the issue of stigma was to dispense PrEP at drop-in centers, where MSM clients would feel more comfortable visiting "because there is more privacy".

## Discussion

Our study explored PrEP knowledge and acceptability and barriers and facilitators of its implementation among MSM, trans women, and other GDSM and HIV providers and stakeholders in Ghana. Our findings indicate that while most MSM, trans women, and GDSM clients were unaware of PrEP, they, along with providers/stakeholders, were overwhelmingly interested in PrEP use and implementation for MSM in Ghana to reduce risk of HIV transmission. Our findings on medical, social behavioral, and structural concerns about PrEP uptake and implementation reveal important insights on gaps in MSM, trans women, and GDSM's and providers' PrEP knowledge, as well as key multilevel barriers to its use and roll-out among MSM, trans women, and GDSM. Notably underlying many of the concerns and barriers of PrEP use were MSM, trans women, and GDSM's experiences or anticipation of anti-gay and HIV stigma and its impact on PrEP implementation and uptake among the group. Moreover, some providers and stakeholders shared stigmatizing views about MSM that reflect the intersectional stigma MSM, trans women, and GDSM report and anticipate facing in the healthcare setting. Our findings highlight a challenge that intersectional stigma and its internalization are both a core reason for MSM, trans women, and GDSM interest in PrEP *and yet* a barrier for uptake. These findings can inform Ghana's newly developed PrEP program, including the implementation of targeted educational interventions to address PrEP knowledge gaps and inform proper use of PrEP, as well as interventions to reduce intersectional stigma and other barriers to PrEP access, use, and implementation for MSM, trans women, and GDSM.

Our study noted that although all service providers and stakeholders were knowledgeable of PrEP, MSM, trans women, and GDSM clients had low knowledge of it, similar to another study in Ghana [18] and other West African countries [17], but also lower than many other countries outside the region [21, 22]. The higher knowledge in other sub-Saharan African regions may be because PrEP programming on the continent has primarily focused on East and Southern Africa, where the HIV epidemic is more generalized. However, clients, along with providers/stakeholders, expressed great interest in PrEP use among MSM, trans women, and GDSM and agreed that PrEP roll-out should be prioritized in Ghana, especially for serodiscordant couples and at-risk MSM, reflecting findings from other studies in Africa [17, 18, 22, 31].

A key finding is that many of the underlying reasons MSM, trans women, and GDSM gave for their interest in PrEP use was related to intersectional anti-gay and HIV stigma and how HIV has compromised their social freedom due to these stigmas and the association of gay men with HIV. This suggests that PrEP access and use may empower MSM, trans women, and GDSM in managing HIV risk and mitigating the negative impact of intersectional stigma on their lives. We also found that similar to other studies of MSM and trans women in Africa [19, 20, 22, 32], some clients viewed PrEP use as a way to reduce risk for HIV during riskier sexual encounters (i.e., group sex or condomless anal intercourse) or in circumstances where they are unaware of their sexual partner's HIV status, suggesting that condom use or HIV status disclosure may be a difficult in sexual intimacies. Risk perception has been found to be a motivating factor in PrEP uptake among MSM [32, 33]. PrEP programming should thus help MSM, trans women, and GDSM assess their risk of HIV to increase PrEP initiation. Participants also

wanted to take PrEP in order to safely have condomless sex as they viewed it as more intimate and pleasureful, similar to other studies [34]. This view suggested a lack of understanding that STIs remain a risk while using PrEP—a key issue in PrEP programming [35]. Thus, education on how PrEP works will be critically important for proper use of it.

Similar to clients, providers and stakeholders viewed PrEP as a way to protect against HIV risk among MSM and expressed frustration with Ghana's delay in introducing PrEP to key populations, with some characterizing the lack of PrEP programming as unethical given PrEP's efficacy in preventing HIV transmission. A key issue, however, is that some provider and stakeholder commentary on HIV risk among MSM and why PrEP was needed reflected stigmatizing views that characterized MSM as promiscuous. Such views contribute to the climate of intersectional anti-gay and HIV stigma that prevent MSM, trans women, and GDSM from seeking HIV services [23] and should be addressed in routine anti-stigma trainings of providers and stakeholders.

Although there was great interest in PrEP, clients and providers/stakeholders shared various overlapping as well as different perspectives and concerns about PrEP use and implementation in Ghana for MSM. Perspectives on barriers and facilitators of PrEP implementation raised ranged from medical concerns (e.g., STIs; drug resistance); social behavioral concerns (e.g., stigma, risk compensation, adherence issues); and structural barriers (e.g., cost/affordability, government commitment, monitoring systems, policy guidance). Both clients and providers/stakeholders emphasized the importance of making PrEP free or affordable to clients due to the socioeconomic conditions facing MSM, trans women, and GDSM—an issue that was raised in a previous research on PrEP in Africa [31] and among MSM in Ghana [18].

For clients, their interest in and ability to use PrEP were also shaped by its accessibility and ease of use (e.g., pill dosing and side effects); intersectional stigma; sexual behavioral preferences (e.g., condomless sex vs. condom use), and HIV risk perception. Many clients, regardless of their interest in PrEP, feared PrEP side effects—a fear that was rooted in their own experiences with ART side effects. Relatedly, some clients expressed concerns about the health risks of PrEP use. These medication safety fears echo some of the concerns raised by other key populations about PrEP uptake [14, 17, 24, 36, 37]. The fear of possible side effects and health risks are important to address in local efforts to increase PrEP uptake given that side effects are a commonly cited issue by those who discontinue using PrEP [32]. In fact, a preliminary report on Ghana's PrEP rollout indicates that key populations taking PrEP discontinue due to side effects and pill fatigue [25]. PrEP side effects are mild, affect a minority of people, and subside within a month of dosing [38]. Clients should be educated with this information and monitored closely in the first month of uptake to minimize discontinuation.

Due to their roles within the healthcare system, an issue providers and stakeholders emphasized more than clients was health system challenges such as the country's weak monitoring/surveillance system that may affect provider's ability to monitor PrEP enrollment and outcomes, as well as concerns about potential PrEP shortages due to a history of antiretroviral shortages in Ghana [39]. They also raised more social behavioral and related medical concerns than clients. Similar to concerns raised by providers in other contexts about PrEP use among MSM [24], providers and KIs in our study feared antiretroviral resistance among clients after seroconversion or due to poor adherence to PrEP, and risk compensation resulting in STI infections, which clients did not discuss.

Stigma, particularly intersectional HIV and anti-gay stigma, was a more prominent theme among MSM, trans women, and GDSM than providers in concerns shared about PrEP uptake and implementation, reflecting findings from research on PrEP use among Black MSM in the U.S. [40]. For example, some MSM, trans women, and GDSM clients feared that taking a daily pill would stir rumors, a concern other PrEP research among African MSM and trans women

has found [19, 32], or would stigmatize them as ill, a common concern in the HIV literature [41]. While PrEP is a prophylaxis, participants' concerns suggest that pills are socially perceived in Ghana as medication and medication as what one takes to treat illness and thus, a source of stigma. The signification of PrEP dosing as a sign of illness and a catalyst for experiencing stigma in contexts like Ghana must be addressed in PrEP programming as this concern may lower PrEP uptake. Some participants also feared that visiting a clinic for PrEP would out them to clinic staff and attendees as MSM, trans women, or GDSM—a concern also found in research on MSM's willingness to use PrEP in low and middle-income countries [22]. Providers and stakeholders were similarly concerned that the stigmatizing healthcare setting for clients would hinder them from seeking PrEP and that this may create a black market for PrEP use, a concern with implications on the ability of providers to monitor its use.

## Recommendations

Our findings offer several recommendations that can help inform Ghana's new roll out of PrEP. In terms of social behavioral interventions, there is a need for educational campaigns that inform the public about what PrEP is, how to use it, and ways to access it in Ghana. Although PrEP is prioritized for high-risk groups, public campaigns are a way to reach broader populations, including hard to reach groups. Such campaigns can also help reduce stigma around HIV and PrEP use by educating the public about advances in HIV prevention and normalizing public discussions of HIV. PrEP programming should also include comprehensive education on the proper use of PrEP due to medical concerns providers shared about improper use of PrEP and risk compensation. Such a program should also include information on PrEP side effects and provider support on managing these.

For those who fear medication side effects, stigma of accessing PrEP at clinics, or daily pill use, a potential option is intermittent/on-demand dosing of PrEP around time of sexual engagement, which has been found to reduce risk of HIV incidence by 86% among high-risk men [42]. Stakeholders should consider including on-demand dosing as part of their PrEP programming, including a strong educational component on how to intermittently take PrEP correctly and the risks of this dosing option. In addition, injectable PrEP, which was approved by FDA in December 2021 [43], should be offered in contexts like Ghana without delay for its ease of use, to facilitate adherence, and to help address Ghanaian MSM's concerns about the stigmas associated with pill-dosing. Offering these options could help retain those who fear pill fatigue and side effects—factors that have been cited as the primary reasons for discontinuation of PrEP use in Ghana [25].

Finally, in terms of health systems/structural level recommendations, easy and confidential access to PrEP in stigma-free settings will be key to whether MSM, trans women, and GDSM enroll on PrEP. Concerns clients shared about intersectional stigma, combined with our observation of stigmatizing views among providers of MSM's sexualities, raise an urgent need for ongoing intersectional anti-stigma training among providers and a strengthening of confidential HIV service provision to ensure respect of and professionalism in engagement with MSM, trans women, and GDSM. The stakes are particularly high in Ghana given the context of criminalization of male same-sex sex and ongoing efforts to pass a bill that would criminalize LGBTQI people, advocacy, and services [44]. Thus, attention to confidentiality and stigma will be key to clients feeling safe and comfortable to access PrEP services. Additionally, to ensure there are no economic barriers to accessing PrEP, it should be offered for free as should all laboratory tests needed for PrEP initiation. An assessment of the country's rollout found that at sites that required payment for PrEP initiation laboratory tests, many clients could not afford the fees [25]. Clear guidelines and trainings of service providers on how to properly prescribe

PrEP, counsel clients, and monitor PrEP use would also be critical to successful PrEP implementation and proper use. This is especially important given provider and KI perspectives that suggested they were confusing PEP and PrEP.

## Limitations

There are a few limitations to our study. First, our study utilized purposive sampling, thus, findings cannot be generalized to all MSM, trans women, GDSM, providers, or stakeholders in Ghana. Second, we conducted interviews with only English-speaking participants, thus our findings may not be generalizable to non-English speakers, though a majority of Ghanaians speak English. Third, due to the onset of the COVID-19 pandemic, we were unable to conduct member checking to validate study findings. Finally, our study asked MSM, trans women, and GDSM who are already living with HIV about their views on PrEP. We recognize that their responses may be biased by hindsight; nevertheless, we thought it would be important to field questions with them about PrEP knowledge, interest, barriers, and facilitators of PrEP use since PrEP implementation will be most critical for individuals at high risk of seroconversion. However, a future study examining the same questions among serodiscordant MSM, trans women, and GDSM partners and with HIV-negative MSM, trans women, and GDSM would offer important insights on PrEP use among key populations. Additionally, a study with current PrEP users in Ghana would help identify experienced barriers and facilitators of PrEP use.

## Conclusion

Our study suggests that despite low knowledge of PrEP among MSM, trans women, and GDSM in Ghana, they, along with providers and stakeholders, have great interest in PrEP availability and use. However, there are various multilevel factors that may hinder optimal PrEP uptake among MSM, trans women, and GDSM. Some of these barriers, like fear of side effects and prohibitive lab fees, have already hindered uptake in the first year of PrEP rollout in Ghana. PrEP availability is critical for MSM and GDSM in Ghana, especially given the high rates of sexual violence they experience [10] and the restrictive and stigmatizing social and legal context that criminalize same-sex sexualities and increases HIV risk. The country's recent implementation of PrEP holds great promise in reducing HIV incidence among MSM and GDSM; however, key stakeholders must ensure PrEP is rolled out with a strong educational component and that it is affordable and accessible. Moreover, and critically important, intersectional stigma must be addressed for PrEP use to reach its potential, freeing MSM, trans women, and GDSM to access user-friendly HIV services.

## Acknowledgments

We would like to thank study participants for their time and contributions to the study and service providers for their support with recruitment.

## Author Contributions

**Conceptualization:** Akua O. Gyamerah, Sheri A. Lippman.

**Data curation:** Akua O. Gyamerah, Guro Sorensen, Matilda D. Mensah, Naa Ashiley Vanderpuye.

**Formal analysis:** Akua O. Gyamerah, Ezra Kinzer.

**Funding acquisition:** Akua O. Gyamerah.

**Investigation:** Akua O. Gyamerah.

**Methodology:** Akua O. Gyamerah.

**Supervision:** Akua O. Gyamerah, Naa Ashiley Vanderpuye, Sheri A. Lippman.

**Writing – original draft:** Akua O. Gyamerah, Ezra Kinzer, Gloria Aidoo-Frimpong.

**Writing – review & editing:** Akua O. Gyamerah, Guro Sorensen, Matilda D. Mensah, Kelly D. Taylor, Naa Ashiley Vanderpuye, Sheri A. Lippman.

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
