## [Decision Letter · Decision Letter 0]

5 Jan 2023

PGPH-D-22-01670

PrEP knowledge, acceptability, and implementation in Ghana: Perspectives of HIV service providers and MSM living with HIV

Dear Dr. Gyamerah,

Thank you for submitting your manuscript to PLOS Global Public Health. After careful consideration, we feel that it has merit but does not fully meet PLOS Global Public Health’s publication criteria as it currently stands. Therefore, we invite you to submit a revised version of the manuscript that addresses the points raised during the review process.

Please note that we have only been able to secure a single reviewer to assess your manuscript. We are issuing a decision on your manuscript at this point to prevent further delays in the evaluation of your manuscript. Please be aware that the editor who handles your revised manuscript might find it necessary to invite additional reviewers to assess this work once the revised manuscript is submitted. However, we will aim to proceed on the basis of this single review if possible. 

 The reviewer has raised a number of concerns that need attention. Could you please revise the manuscript to carefully address the concerns raised? 

We look forward to receiving your revised manuscript.

Kind regards,

Vanessa Carels

Staff Editor

Journal Requirements:

1. Please amend your Data Availability Statement and indicate where the data may be found.

Additional Editor Comments (if provided):

Reviewers' comments:

Reviewer's Responses to Questions

**Comments to the Author**

1. Does this manuscript meet PLOS Global Public Health’s publication criteria? Is the manuscript technically sound, and do the data support the conclusions? The manuscript must describe methodologically and ethically rigorous research with conclusions that are appropriately drawn based on the data presented.

Reviewer #1: Yes

2. Has the statistical analysis been performed appropriately and rigorously?

Reviewer #1: N/A

3. Have the authors made all data underlying the findings in their manuscript fully available (please refer to the Data Availability Statement at the start of the manuscript PDF file)?

Reviewer #1: No

4. Is the manuscript presented in an intelligible fashion and written in standard English?

Reviewer #1: Yes

5. Review Comments to the Author

Reviewer #1: This is a really interesting, well written and insightful article about PrEP in Ghana. I was initially concerned by the research having been conducted primarily with individuals who have HIV and were therefore not PrEP candidates, however I feel the authors addressed these concerns well and presented a compelling case for why this research matters. In addition, the introduction and methods were very strong. I have some minor comments which I think will strengthen the manuscript.

1) I understand the framing around the sample as coming under the umbrella of MSM but given that trans women were included I do not think it appropriate that it be described as such. Instead the authors might consider MSM (cis and trans) and gender diverse people or something similar (especially if trans men were included as well as trans women).

This is especially important given that such a high proportion have female reported for gender (n=8) and given the inclusion criteria which includes those assigned male at birth or identifying as men. This would benefit from elaboration as it isn’t clear if this is the sex (gender) assigned at birth and these individuals are trans men, or if a number of participants assigned male at birth identified as female and other as trans women.

Essentially, as it stands it is difficult to understand the sample composition and the label of MSM does not help the reader understand and therefore weakens the article.

2) It is helpful that the authors describe recruiting people who have HIV as a limitation. It is worth stating here that it is critical that further research is conducted with a range of candidates for or user of PrEP in Ghana, rather than just partners in sero-different relationships.

3) It is worth including demographic details alongside participant quotes. It is a lot of work for the reader to have to refer back to the tables to contextualise the quotes.

Thank you for the opportunity to review this interesting work.

6. PLOS authors have the option to publish the peer review history of their article (what does this mean?). If published, this will include your full peer review and any attached files.

**Do you want your identity to be public for this peer review?** For information about this choice, including consent withdrawal, please see our Privacy Policy.

Reviewer #1: No

---

## [Decision Letter · Decision Letter 1]

2 May 2023

PrEP knowledge, acceptability, and implementation in Ghana: Perspectives of HIV service providers and MSM, trans women, and gender diverse individuals living with HIV

PGPH-D-22-01670R1

Dear Dr. Gyamerah,

We are pleased to inform you that your manuscript 'PrEP knowledge, acceptability, and implementation in Ghana: Perspectives of HIV service providers and MSM, trans women, and gender diverse individuals living with HIV' has been provisionally accepted for publication in PLOS Global Public Health.

Best regards,

Siyan Yi, MD, MHSc, PhD

Academic Editor

Reviewer Comments (if any, and for reference):

Reviewer's Responses to Questions

**Comments to the Author**

1. If the authors have adequately addressed your comments raised in a previous round of review and you feel that this manuscript is now acceptable for publication, you may indicate that here to bypass the “Comments to the Author” section, enter your conflict of interest statement in the “Confidential to Editor” section, and submit your "Accept" recommendation.

Reviewer #1: All comments have been addressed

2. Does this manuscript meet PLOS Global Public Health’s publication criteria? Is the manuscript technically sound, and do the data support the conclusions? The manuscript must describe methodologically and ethically rigorous research with conclusions that are appropriately drawn based on the data presented.

Reviewer #1: Yes

3. Has the statistical analysis been performed appropriately and rigorously?

Reviewer #1: N/A

4. Have the authors made all data underlying the findings in their manuscript fully available (please refer to the Data Availability Statement at the start of the manuscript PDF file)?

Reviewer #1: Yes

5. Is the manuscript presented in an intelligible fashion and written in standard English?

Reviewer #1: Yes

6. Review Comments to the Author

Reviewer #1: Well done on an excellent paper! I look forward to seeing it published.

7. PLOS authors have the option to publish the peer review history of their article (what does this mean?). If published, this will include your full peer review and any attached files.

**Do you want your identity to be public for this peer review?** For information about this choice, including consent withdrawal, please see our Privacy Policy.

Reviewer #1: No
